# A Lateral-Flow Device for the Rapid Detection of *Scedosporium* Species

**DOI:** 10.3390/diagnostics14080847

**Published:** 2024-04-19

**Authors:** Genna E. Davies, Christopher R. Thornton

**Affiliations:** 1ISCA Diagnostics Ltd., B12A, Hatherly Laboratories, Prince of Wales Road, Exeter EX4 4PS, UK; g.davies@exeter.ac.uk; 2Biosciences, Faculty of Health and Life Sciences, Prince of Wales Road, Exeter EX4 4PS, UK

**Keywords:** *Scedosporium*, scedosporiosis, monoclonal antibody, biomarker, lateral-flow device, enzyme-linked immunosorbent assay

## Abstract

*Scedosporium* species are human pathogenic fungi, responsible for chronic, localised, and life-threatening disseminated infections in both immunocompetent and immunocompromised individuals. The diagnosis of *Scedosporium* infections currently relies on non-specific CT, lengthy and insensitive culture from invasive biopsy, and the time-consuming histopathology of tissue samples. At present, there are no rapid antigen tests that detect *Scedosporium*-specific biomarkers. Here, we report the development of a rapid (30 min) and sensitive (pmol/L sensitivity) lateral-flow device (LFD) test, incorporating a *Scedosporium*-specific IgG1 monoclonal antibody (mAb), HG12, which binds to extracellular polysaccharide (EPS) antigens between ~15 kDa and 250 kDa secreted during the hyphal growth of the pathogens. The test is compatible with human serum and allows for the detection of the *Scedosporium* species most frequently reported as agents of human disease (*Scedosporium apiospermum*, *Scedosporium aurantiacum*, and *Scedosporium boydii*), with limits of detection (LODs) of the EPS biomarkers in human serum of ~0.81 ng/mL (*S. apiospermum*), ~0.94 ng/mL (*S. aurantiacum*), and ~1.95 ng/mL (*S. boydii*). The *Scedosporium*-specific LFD (*Sced*LFD) test therefore provides a potential novel opportunity for the detection of infections caused by different *Scedosporium* species.

## 1. Introduction

*Scedosporium* species are human pathogenic moulds. They are the agents of eumycetoma [1,2,3,4,5], a chronic deep fungal infection of the skin and subcutaneous tissues, and are responsible for a broad spectrum of localised and life-threatening disseminated infections in immunocompetent and immunocompromised individuals [6,7,8,9,10] affecting numerous organs of the body (Figure 1A), including the bones and joints [11,12,13], the central nervous system [14,15,16,17], the eyes [18,19,20,21,22,23,24], the lungs [14,25,26,27,28,29,30,31,32,33], the sinuses [34,35,36], and other body sites [37,38,39] of cystic fibrosis patients [8,28,40,41,42], haematopoietic stem cell and solid organ transplant recipients [31,43,44,45,46,47], hospitalised patients with COVID-19 [48], victims of near-drowning following natural disasters [14,15,30,49,50], and persons with traumatic injuries [51,52,53,54,55]. Recently assigned to the high- (eumycetoma-causative agents) and moderate-priority pathogen groupings by the World Health Organisation [56], *Scedosporium* diseases have an overall all-cause mortality rate of 42–46% [56].

The *Scedosporium* species most commonly associated with life-threatening infections are *S. apiospermum*, *S. aurantiacum*, and *S. boydii* [8,57,58,59,60,61]. The detection of these pathogens relies on a combination of techniques including the histological examination and *in situ* hybridisation of tissue samples [62], isolation of the fungi from biopsy samples using semi-selective or selective media, and microscopical examination of cultures for characteristic morphological features [58,63], with species identification requiring matrix-assisted laser desorption/ionisation–time-of-flight mass spectrometry (MALDI-TOF MS) [64] or molecular methods such as polymerase chain reaction [58,63]. While immunoassays that employ patient sera have been developed for the detection of *Scedosporium* and *Lomentospora* species in cystic fibrosis patients [65,66], there is no rapid antigen test currently available for the specific detection of *Scedosporium* species [63].

Here, we report the development of a lateral-flow device (*Sced*LFD) for the rapid (30 min) and sensitive (pmol/L) detection of a *Scedosporium*-specific biomarker. The immunoassay employs a *Scedosporium*-specific monoclonal antibody (mAb, HG12) which binds to extracellular polysaccharide (EPS) antigens present on the spore and hyphal cell wall of these fungi [67] and which is secreted during hyphal growth. The *Sced*LFD test is compatible with human serum, with limits of detection of EPS antigens in serum of ~0.81 ng/mL (*S. apiospermum*), ~0.94 ng/mL (*S. aurantiacum*), and ~1.95 ng/mL (*S. boydii*). The LFD therefore provides a novel opportunity for the rapid, sensitive, and specific detection of these human pathogenic fungi.

## 2. Materials and Methods

### 2.1. Monoclonal Antibody

The mouse monoclonal antibody (mAb) HG12 [67] is an immunoglobulin G1 (IgG1) κ-light chain antibody and is specific to species in the *Scedosporium apiospermum* complex (*Scedosporium angustum*, *Scedosporium apiospermum*, *Scedosporium boydii*, *Scedosporium ellipsoideum*, and *Scedosporium fusoideum*), *Scedosporium aurantiacum*, *Scedosporium desertorum*, and *Scedosporium minutisporum*. The mAb also recognises the related fungi *Parascedosporium tectonae*, *Petriellopsis africana*, *Lophotrichus fimeti*, *Graphium eumorphum* (the *Graphium* type of *S. apiospermum*), and the teleomorph form of *Graphium*, *Petriella setifera*. The mAb does not react with *Scedosporium dehoogii* or with unrelated moulds and yeasts of clinical importance including *Lomentospora prolificans* (formerly *Scedosporium prolificans*).

### 2.2. Fungal Culture

Fungi (Table 1) were routinely cultured on oatmeal agar (OA; P2182, Sigma, Gillingham, UK) or malt extract agar (MEA; 70145, Sigma). Media were autoclaved at 121 °C for 15 min prior to use, and fungi were grown at 37 °C. Extracellular polysaccharides (EPSs) were purified from 6 d old culture filtrates of fungi grown at 30 °C in YNB+G liquid medium as described previously [68]. For *Sced*LFD and *Sced*ELISA specificity tests, 72 h old YNB+G culture filtrates of fungi [69] were used.

### 2.3. Antibody Purification and Enzyme Conjugation

The hybridoma tissue culture supernatant of mAb HG12 was harvested by centrifugation at 2147× *g* for 40 min at 4 °C, followed by filtration through a 0.8 μM cellulose acetate filter (10462240, GE Healthcare Life Sciences, Amersham, UK). The culture supernatant was loaded onto a HiTrap Protein A column (17-0402-01, GE Healthcare Life Sciences) using a peristaltic pump P-1 (18-1110-91, GE Healthcare Life Sciences) with a low pulsation flow of 1 mL/min. Columns were equilibrated with 10 mL of phosphate-buffered saline (PBS; PBS; 137 mM NaCl, 2.7 mM KCl, 8 mM Na_2_HPO_4_, 1.5 mM KH_2_PO_4_ [pH 7.2]), and column-bound antibody was eluted with 5 mL of 0.1 M glycine–HCl buffer (pH 2.5) with a flow rate of 0.5 mL/min. The buffer of the purified antibody was exchanged to PBS using a disposable PD-10 desalting column (17-0851-01, GE Healthcare Life Sciences). Following purification, the antibody was sterile filtered with a 0.24 µm syringe filter (85037-574-44, Sartorius, Epsom, UK) and stored at 4 °C. Antibody purity was confirmed by SDS-PAGE and gel staining using Coomassie Brilliant Blue R-250 dye (20278, Thermo Fisher Scientific, Loughborough, UK). Protein A-purified mAb HG12 was conjugated to horseradish peroxidase (HRP) for ELISA studies using a Lightning-Link horseradish peroxidase conjugation kit (701-0000; Bio-Techne Ltd., Abingdon, UK), or to alkaline phosphatase (AKP) for Western blotting studies using a Lightning-Link alkaline phosphatase conjugation kit (702-0010; Bio-Techne Ltd.).

### 2.4. Polyacrylamide Gel Electrophoresis and Western Blotting

Sodium dodecyl sulphate–polyacrylamide gel electrophoresis (SDS-PAGE) was carried out using 4–20% gradient polyacrylamide gels (161-1159, Bio-Rad, Watford, UK) under denaturing conditions. Antigens were separated electrophoretically at 165 V, and pre-stained markers (161-0318, Bio-Rad) were used for molecular weight determinations. For Western blotting, separated antigens were transferred electrophoretically onto a PVDF membrane (162-0175, Bio-Rad) for 2 h at 75 V, and the membrane was blocked for 16 h at 4 °C in PBS containing 1% (wt:vol) BSA. Blocked membranes were incubated with HG12-AKP conjugate diluted 1:15,000 (vol:vol) in PBS containing 0.5% (wt:vol) BSA (PBSA) for 2 h at 23 °C. Membranes were washed three times with PBS, once with PBST, and bound antibody visualised by incubation in a substrate solution. Reactions were stopped by immersing membranes in dH_2_O, and membranes were then air-dried between sheets of Whatman filter paper.

### 2.5. LFD and ELISA Specificities

#### 2.5.1. Lateral-Flow Device

The *Scedosporium* lateral-flow device (*Sced*LFD) was manufactured by Lateral Dx (Alloa, Scotland, UK). The test consists of a Sartorius CN95 nitrocellulose (NC) membrane laminated with an absorbent pad, sample pad, and LDX-treated polyester conjugate pad containing 2.5 μL of Protein A-purified mAb HG12 conjugated to RE1 red cellulose nanobeads (Asahi Kasei, Tokyo, Japan). The NC test (T) line consists of Protein A-purified mAb HG12 at a concentration of 2 mg/mL, while the internal test control (C) line consists of goat anti-mouse IgG (Arista Biologicals, Allentown, PA, USA) at a concentration of 1 mg/mL. For specificity tests, 72 h old culture filtrate was mixed 1:10 (vol:vol) with LFD running buffer (RB; PBS containing 0.1% (vol:vol) Tween-20), and 100 μL of the resultant solution was added to the LFD test. The negative control for *Sced*LFD tests consisted of YNB+G medium only diluted 1:10 (vol:vol) with RB. After 30 min, the intensities of the test (T) and control (C) lines were determined as artificial units (a.u.) using a Cube reader [68,69].

#### 2.5.2. Sandwich Enzyme-Linked Immunosorbent Assay

For the *Scedosporium* sandwich ELISA (*Sced*ELISA), wells of Maxisorp microtitre plates (10547781, Thermo Fisher Scientific, Loughborough, UK) were coated with 50 µL volumes of Protein A-purified mAb HG12 at a concentration of 3 mg/mL PBS. After incubation for 16 h at 4 °C, the wells were washed three times (5 min each wash) with PBST (PBS containing 0.05% (vol:vol) Tween-20), once with PBS for 5 min, and then given a final rinse with dH_2_O before air-drying at 23 °C. Antibody-coated wells were incubated at 23 °C for 1 h with 50 µL of 72 h old culture filtrates diluted 1:10 (vol:vol) with PBST (control wells incubated with YNB+G medium only diluted 1:10 (vol:vol) with PBST), after which they were given four 5 min washes with PBST. Washed wells were then incubated for 1 h at 23 °C with HG12-HRP conjugate diluted 1 in 5000 (vol:vol) in PBST, after which they were washed four times with PBST as described, given a final 5 min wash with PBS, and bound antibody visualised by incubating wells with tetramethyl benzidine (TMB) substrate solution for 30 min. Enzyme–substrate reactions were stopped by the addition of 3 M H_2_SO_4_, and absorbance values were determined at 450 nm using a microplate reader (infinite F50, Tecan Austria GmbH, Reading, UK). All incubation steps were performed at 23 °C in sealed plastic bags.

### 2.6. Serological Detection and Limits of Detection

Normal serum from healthy AB blood group males (H6914, Sigma) was spiked with purified EPS from *S. apiospermum* isolate CBS8353, *S. aurantiacum* isolate CBS121926, and *S. boydii* isolate CBS835.96 and stored as aliquots at −20 °C prior to use. The standard operating procedure (SOP) for serum pre-treatment and testing using the *Sced*LFD test is illustrated in Figure 1B. For the thawing of serum, 50 μL spiked or control (unspiked) serum was mixed 1:2 (vol:vol) with PBS containing 0.5% (wt:vol) Na_2_-EDTA (pH6.0) and heated in a boiling water bath for 3 min. The heated mixture was centrifuged at 16,000× *g* for 5 min, the clear supernatant mixed 1:1 (vol:vol) with LFD RB, and the resultant solution assayed by *Sced*LFD (100 μL per test) as described. For testing by *Sced*ELISA (50 μL per well), supernatants were diluted in PBST and assayed as described but with an antigen incubation step of 2 h.

### 2.7. Statistical Analysis

Numerical data were analysed using Student’s *t*-test (independent, two-tailed) to determine statistical significance.

## 3. Results

### 3.1. Lateral-Flow Device and ELISA

#### 3.1.1. Specificities

Using culture filtrates from yeasts and moulds grown for 72 h in YNB+G liquid medium, both the *Sced*LFD and the *Sced*ELISA were shown to be *Scedosporium*-specific (Table 1), reacting strongly with filtrates from species in the *Scedosporium apiospermum* complex (*Scedosporium angustum*, *Scedosporium apiospermum*, *Scedosporium boydii*, *Scedosporium ellipsoideum*, and *Scedosporium fusoideum*) and with filtrates from *Petriella setifera*, *Scedosporium aurantiacum*, *Scedosporium desertorum*, and *Scedosporium minutisporum*. Neither of the immunoassays cross-reacted with filtrates from unrelated yeasts or moulds of clinical importance including *Candida albicans*, *Cryptococcus neoformans*, *Aspergillus* spp., *Fusarium* spp., *Mucorales* fungi (species of *Apophysomyces*, *Cunninghamella*, *Lichtheimia*, *Mucor*, and *Rhizopus*), and *Lomentospora prolificans*.

#### 3.1.2. Limits of Detection with Human Serum

Both the *ScedLFD* and the *ScedELISA* are compatible with human serum. In Western blotting studies of extracellular polysaccharides (EPSs) purified from culture filtrates of *Scedosporium apiospermum*, *Scedosporium aurantiacum*, and *Scedosporium boydii*, mAb HG12 reacted strongly with antigens with molecular weights between ~15 kDa and 250 kDa but did not cross-react with EPS antigen purified from *Aspergillus fumigatus* culture filtrate (Figure 2). Using the purified EPS samples, *Sced*LFD was shown to have limits of detection of ~0.26 ng/mL running buffer (RB), ~0.24 ng/mL RB, and ~0.98 ng/mL RB for *S. aurantiacum* (Figure 3A), *S. apiospermum* (Figure 3C), and *S. boydii* (Figure 3E), respectively. In serum tests, *Sced*LFD showed limits of detection of ~0.94 ng/mL serum, ~0.81 ng/mL serum, and ~1.95 ng/mL serum for *S. aurantiacum* (Figure 3B), *S. apiospermum* (Figure 3D), and *S. boydii* (Figure 3F), respectively. In serum tests, *Sced*ELISA was less sensitive than the *Sced*LFD test, with limits of detection of ~15.6 ng/mL serum, ~62.5 ng/mL serum, and ~125 ng/mL serum for *S. aurantiacum*, *S. apiospermum*, and *S. boydii*, respectively (Figure 4).

## 4. Discussion

In this paper, we describe the development of a lateral-flow device (*Sced*LFD) test for the rapid detection of *Scedosporium* species, fungal pathogens responsible for myriad chronic and life-threatening infections of the skin and subcutaneous tissues (eumycetoma), bones and joints, central nervous system, sinuses, eyes, and lungs of humans (Figure 1A).

*Sced*LFD incorporates a monoclonal antibody (mAb), HG12, previously raised against *Scedosporium boydii* [67] and which is specific, recognising species in the *Scedosporium apiospermum* complex (*Scedosporium angustum*, *Scedosporium apiospermum*, *Scedosporium boydii*, *Scedosporium ellipsoideum*, and *Scedosporium fusoideum*) and also *Scedosporium aurantiacum*, *Scedosporium desertorum*, and *Scedosporium minutisporum*. In addition, mAb HG12 recognises *Graphium* and *Petriella* states of *Scedosporium*, also reported as human pathogens [20,37,70]. Importantly, the mAb does not cross-react with unrelated yeasts and moulds of clinical importance including *Aspergillus* spp., *Candida* and *Cryptococcus* spp., *Mucorales* spp., and species of *Fusarium*. This high degree of specificity is of critical importance given the occurrence of mixed fungal co-infections in humans [1,19,40,71,72,73,74].

The current detection of infectious *Scedosporium* species relies on sophisticated laboratory tests, including MALDI-TOF MS [64] or molecular methods such as polymerase chain reaction (PCR) [58,63]. While a potential diagnostic antigen (a 50–80 kDa peptidorhamnomannan (PRM)) from *Scedosporium boydii* has been reported [75], and mAbs raised against *Scedosporium apiospermum* PRM, their lack of specificity, cross-reacting with *Candida* spp., *Histoplasma capsulatum*, and *Lomentospora prolificans* [76], limits their use in diagnostic test development. Consequently, this is the first time, to the best of our knowledge, that a mAb-based lateral-flow test has been developed for the specific detection of *Scedosporium* species.

For point-of-care diagnostics employing lateral-flow technology, extracellular antigens are needed that can act as signature molecules of infection [77]. Ideally, these should be produced during the active growth of a pathogen, and the target epitope should be heat-stable allowing for the pre-treatment of biofluids such as serum for biomarker detection [68,69]. The *Scedosporium*-specific mAb HG12 used in the *Sced*LFD test binds to a heat-stable extracellular polysaccharide (EPS) antigen present on the spore and hyphal cell wall [67] and which is secreted during the hyphal growth of *Scedosporium* species (this study). It therefore represents an ideal biomarker for the serological detection of these pathogens.

Using a simple and quick serum pre-treatment method (Figure 1B), we have shown that the *Sced*LFD test is compatible with human serum. As with other sandwich-format LFD tests [78,79], a negative test result is shown by the presence of the control (C) line only (Figure 1C), while a positive test result is shown by the presence of a test (T) line and the C line (Figure 1D). This gain-of-signal at the T line allows for the simple visual appraisal of test positivity but can introduce bias due to the subjective nature of test interpretation. To eliminate bias, the presence and intensity of the T line can be established using a Cube reader, with the test output recorded as artificial units (a.u.). When combined with a Cube reader, the limits of detection of *Sced*LFD for the three most common agents of disease in humans (*S. apiospermum*, *S. aurantiacum*, and *S. boydii*) were shown to be <2 ng EPS/mL serum. This high level of test sensitivity (pmol/L) is consistent with other sandwich LFD tests [80]. We found the lateral-flow format to be more sensitive than an enzyme-linked immunosorbent assay (*Sced*ELISA) using the same mAb, HG12. Furthermore, the speed of the *Sced*LFD test (30 min), compared to *Sced*ELISA (>3 h), adds to its superiority in terms of speed, sensitivity, and ease-of-use and satisfies the ASSURED criteria for diagnostics for the developing world [81]. The *Sced*LFD test is therefore ideally suited to resource-limited settings that lack sophisticated diagnostic facilities and equipment needed to run laboratory-based ELISA, MALD-TOF, or PCR tests.

The compatibility of the *Sced*LFD test (and *Sced*ELISA) with serum means it may be suitable for the detection of disseminated *Scedosporium* infections in humans. It should be noted that the test has yet to be validated in the clinic, but its development using a previously characterised mAb [67] is an important first step towards point-of-care testing for *Scedosporium* diseases. While serum is an appropriate biofluid for the biomarker detection of haematogenous dissemination, other biofluids might serve as better sources of antigenic EPS biomarkers for the rapid detection of sinus, eye, lung, and skin infections, as has been postulated for the point-of-care detection of mucormycosis [82]. Notwithstanding this, the availability of a rapid antigen test for a complex of fungi recently assigned to the high- (eumycetoma-causative agents) and moderate-priority pathogen groupings by the World Health Organisation [56], and its compatibility with a minimally invasive biofluid (serum), makes *Sced*LFD a potentially valuable diagnostic tool for this destructive group of pathogens.

## 5. Conclusions

We have developed a specific, sensitive, and simple lateral-flow immunoassay (*Sced*LFD) for the rapid detection of human pathogenic *Scedosporium* species. The test is compatible with human serum, potentially enabling the accurate and minimally invasive detection of *Scedosporium* species at point-of-care.

## Figures and Tables

**Figure 1 diagnostics-14-00847-f001:**
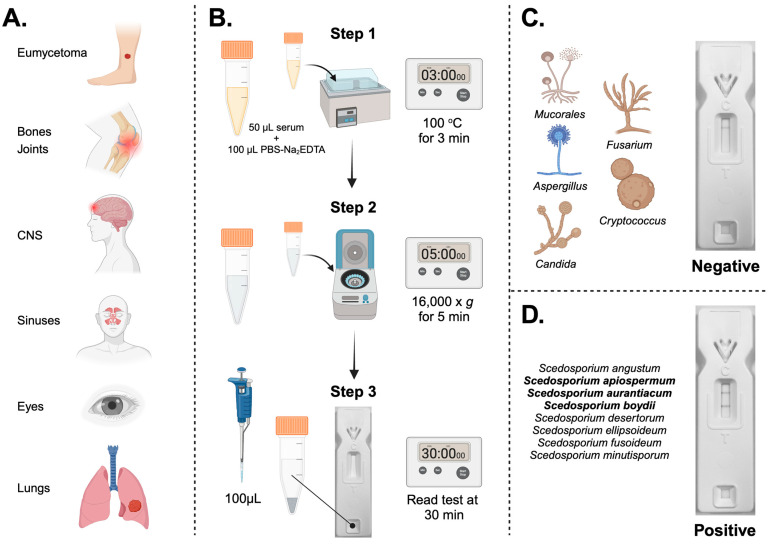
(**A**) The organs of the human body infected by *Scedosporium* spp. Eumycetoma is a chronic deep infection of the skin and subcutaneous tissues. (**B**). The standard operating procedure (SOP) for the treatment of human serum and use of the *Sced*LFD test. Step 1: human serum is mixed 1:2 (vol:vol) with PBS buffer containing Na_2_EDTA and then heated for 3 min at 100 °C in a boiling water bath. Step 2: the heated serum is centrifuged at 16,000× *g* for 5 min to pellet insoluble serum proteins. Step 3: following centrifugation, the clear supernatant is mixed 1:1 (vol:vol) with LFD running buffer, 100 μL is added to the sample port of the LFD test, and after 30 min, the control (C) and test (T) line intensities are determined using a Cube reader. (**C**) A negative test result for unrelated yeasts (*Candida* and *Cryptococcus*) and moulds (*Aspergillus*, *Fusarium*, *Mucorales*) of clinical significance. Note the absence of the test (T) line but the presence of the control (C) line showing that the test has run correctly. (**D**) A positive test result for *Scedosporium* spp. detected by the *Sced*LFD test. The species in bold are responsible for the majority of infections in humans. Note the presence of the test (T) and control (C) lines. Figure ”Created with BioRender.com”.

**Figure 2 diagnostics-14-00847-f002:**
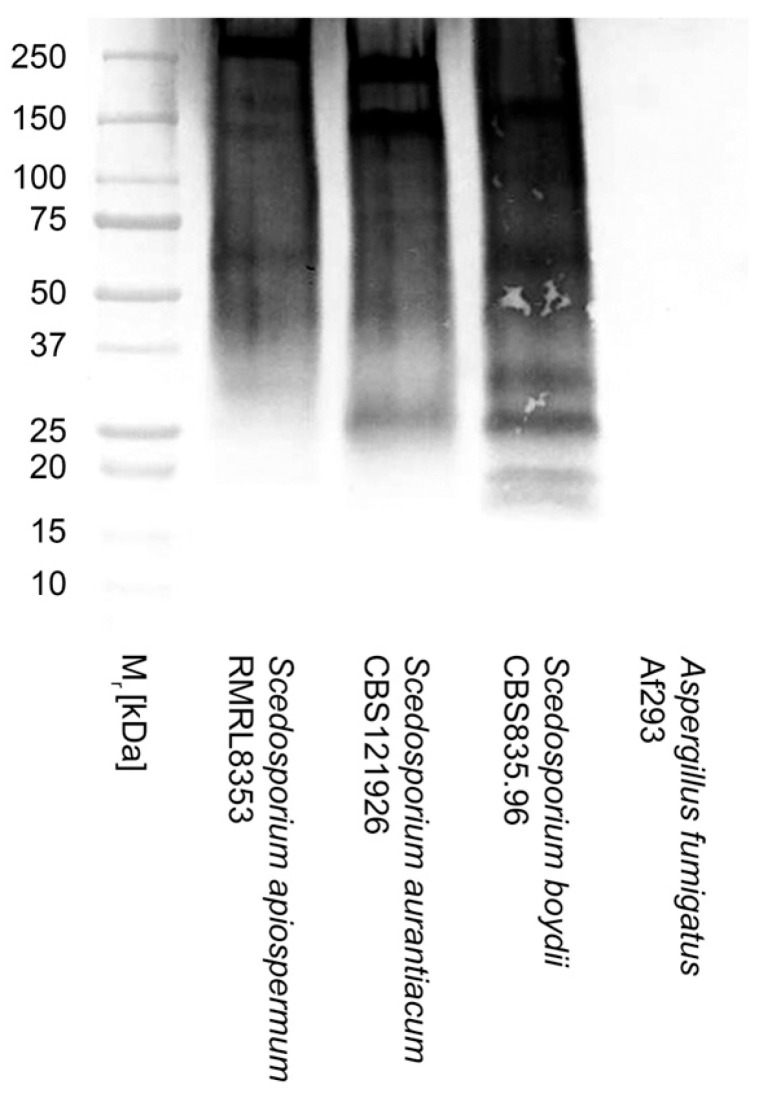
Western blot of EPS antigens (20 μg EPS/lane) from *Scedosporium apiospermum* isolate RMRL8353, *Scedosporium aurantiacum* isolate CBS121926, *Scedosporium boydii* isolate CBS835.96, and *Aspergillus fumigatus* isolate Af293. Note strong binding of mAb HG12 with *Scedosporium* antigens with molecular weights between ~15 kDa and 250 kDa and lack of reactivity with *A. fumigatus* antigens.

**Figure 3 diagnostics-14-00847-f003:**
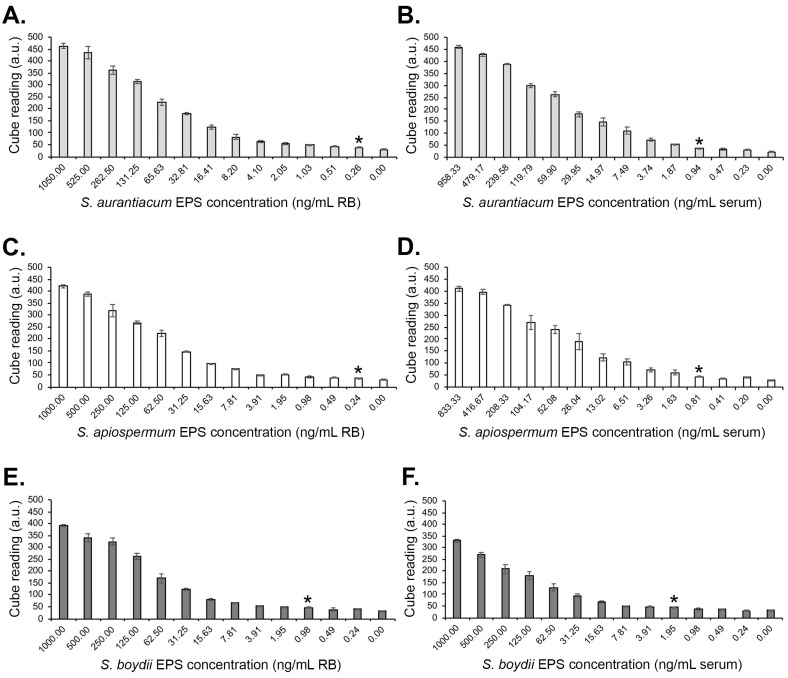
Limits of detection (LODs) of the *Sced*LFD test using purified extracellular polysaccharide (EPS) antigens from *S. aurantiacum* isolate CBS121926 (**A**,**B**), *S. apiospermum* isolate RMRL8353 (**C**,**D**), and *S. boydii* isolate CBS835.96 (**E**,**F**). Cube readings of test (T) line intensities measured as artificial units (a.u) for EPS diluted into LFD running buffer (**A**,**C**,**E**) and for EPS diluted into serum (**B**,**D**,**F**). Data points are the means of 2 replicates ± SE. All *Sced*LFD tests had control (C) line scores of >600 a.u. using the Cube reader. The LODs are indicated by asterisks (*), which show a significant (Student’s *t*-test [*p* < 0.05]) difference in a.u. values compared to control samples (unspiked LFD running buffer (0.00 ng/mL RB) and normal unspiked serum (0.00 ng/mL serum), respectively.

**Figure 4 diagnostics-14-00847-f004:**
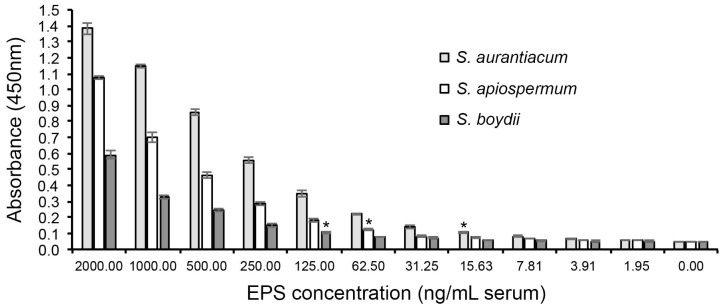
Limits of detection (LODs) in serum of *Sced*ELISA using purified extracellular polysaccharide (EPS) antigens from *S. aurantiacum* isolate CBS121926, *S. apiospermum* isolate RMRL8353, and *S. boydii* isolate CBS835.96. Each data point is the mean of three replicates ± SE. The LODs for each species are indicated by asterisks (*), which show a significant (Student’s *t*-test [*p* < 0.05]) difference in absorbance values compared to matched control samples (normal unspiked serum (0.00 ng/mL serum)). The LODs of *Sced*ELISA are ~15.6 ng/mL serum, ~62.5 ng/mL serum, and ~125 ng/mL serum for *S. aurantiacum*, *S. apiospermum*, and *S. boydii*, respectively.

**Table 1 diagnostics-14-00847-t001:** The details of the fungi used in this study, and the results of *Sced*LFD and *Sced*ELISA tests using culture filtrates from fungi grown as shake cultures for 72 h in YNB+G medium. ^1^ CBS; Westerdijk Fungal Biodiversity Institute, The Netherlands. CRT; C. R. Thornton, University of Exeter, UK. RMRL; Regional Mycology Reference Laboratory, University Hospital, South Manchester, England, UK. FGSC; ^2^ For *Sced*LFD tests, the test (T) line Cube readings in artificial units (a.u.) are the means of two replicate values. The threshold T line value for LFD test positivity is ≥60 a.u. (2 × a.u. value for the YNB+G-only negative control). All LFD tests had control (C) line values of ≥600 a.u. ^3^ For ELISA tests, the absorbance values are the means of two replicate values, and the threshold absorbance value for test positivity is ≥0.100 (2 × absorbance value for the YNB+G-only negative control).

Species	Isolate Number	Source ^1^	*Sced*LFD a.u. ^2^	*Sced*ELISA (Abs 450 nm) ^3^
*Apophysomyces variabilis*	658.93	CBS	28.8	0.052
*Aspergillus fumigatus*	Af293	FGSC	35.7	0.046
*Aspergillus flavus*	144B	CRT	27.9	0.049
*Aspergillus nidulans*	A4	FGSC	35.1	0.044
*Aspergillus niger*	102.4	CBS	32.7	0.052
*Aspergillus terreus* var. *terreus*	601.65	CBS	30.3	0.049
*Candida albicans*	SC5314	SB	37.1	0.060
*Cryptococcus neoformans*	8710	CBS	46.5	0.055
*Cunninghamella bertholletiae*	151.80	CBS	34.5	0.053
*Fusarium oxysporum*	167.3	CBS	34.0	0.052
*Fusarium solani*	224.34	CBS	31.7	0.047
*Lichtheimia corymbifera*	109940	CBS	30.5	0.053
*Lichtheimia hyalospora*	146576	CBS	33.0	0.045
*Lichtheimia ornata*	142195	CBS	33.7	0.050
*Lichtheimia ramosa*	124197	CBS	32.4	0.049
*Lomentospora prolificans*	3.1	CRT	33.0	0.090
*Lomentospora prolificans*	742.96	CBS	41.3	0.048
*Lomentospora prolificans*	100390	CBS	43.0	0.046
*Mucor circinelloides*	123973	CBS	33.8	0.056
*Mucor circinelloides*	124429	CBS	38.2	0.056
*Mucor circinelloides* f. *circinelloides*	120582	CBS	36.7	0.051
*Mucor indicus*	120.08	CBS	34.4	0.044
*Petriella setifera*	109039	CBS	622.3	1.431
*Rhizomucor pusillus*	120586	CBS	31.7	0.051
*Rhizopus arrhizus* var. *arrhizus*	112.07	CBS	37.5	0.056
*Rhizopus arrhizus* var. *delemar*	607.68	CBS	38.8	0.051
*Rhizopus rhizopodiformis*	102277	CBS	32.3	0.050
*Scedosporium angustum*	254.72	CBS	489.4	1.042
*Scedosporium apiospermum*	8353	RMRL	426.2	0.815
*Scedosporium apiospermum*	117407	CBS	469.5	1.337
*Scedosporium aurantiacum*	118934	CBS	691.1	1.403
*Scedosporium aurantiacum*	121926	CBS	737.1	1.427
*Scedosporium boydii*	835.96	CBS	373.4	1.190
*Scedosporium boydii*	100393	CBS	593.5	1.394
*Scedosporium boydii*	100395	CBS	435.4	0.786
*Scedosporium boydii*	100870	CBS	474.3	1.126
*Scedosporium boydii*	Exton 22A	CRT	449.7	1.121
*Scedosporium dehoogii*	117406	CBS	28.1	0.049
*Scedosporium desertorum*	489.72	CBS	500.2	1.310
*Scedosporium ellipsoideum*	438.75	CBS	525.1	1.315
*Scedosporium fusoideum*	106.53	CBS	598.0	1.290
*Scedosporium minutisporum*	116911	CBS	383.3	0.592
YNB+G only	-	-	29.8	0.048

## Data Availability

The data presented in this study are available on request from the corresponding author but are not publicly available due to commercial confidentialities. Monoclonal antibody HG12 and the LFD are available through ISCA Diagnostics Ltd.

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
