# Peer review of "A Lateral-Flow Device for the Rapid Detection of Scedosporium Species"

_diagnostics, 2024, doi:10.3390/diagnostics14080847_

Round 1
Reviewer 1 Report
Comments and Suggestions for Authors
The authors report the development of a lateral-flow device (ScedLFD) for the rapid (30 min) and sensitive (pmol/L) detection of a Scedosporium-specific biomarker. The immunoassay employs a Scedosporium-specific monoclonal antibody (mAb, HG12) which binds to extracellular polysaccharide (EPS) antigens present on the spore and hyphal cell wall of these fungi, and which is secreted during hyphal growth. The ScedLFD test is compatible with human serum, with limits of detection of EPS antigens in serum of ~0.81 ng/mL (S. apiospermum), ~0.94 ng/mL (S. aurantiacum), and ~1.95 ng/mL (S. boydii). This research demonstrated the potential opportunity for the rapid, sensitive and specific detection of these human pathogenic fungi, therefore, I support its publication in Diagnostics, but the authors should resubmit it after a major and carefully revision. Here are some main flaws of this work:
1. Please give experimental results photos of LFD as the authors described.
2. In figure 1B, the time on timer is not consistent with the illustration. In figure 1D, what does the logo in the bottom left corner mean?
3. Please supplement the results of repeated experiments in Table 1.
4. Please improve the image quality of figure 2, and please supplement the results of repeated experiments and statistic results in figure 2.
5. Since the authors wanted to compare the LOD of ScedLFD test and ScedELISA, please unify the format of Figures 3 and 4.
6. There are several critical papers related lateral flow assay should be discussed in the manuscript, such as Trends in Analytical Chemistry, 2024, 117641 and Biosensors 2023, 13 (3), 352.
Comments on the Quality of English LanguageNone
Author Response
We thank Reviewer 1 for their constructive comments. We address their comments in bold below. We highlight relevant changes in yellow in the corrected manuscript attached.
1. Please give experimental results photos of LFD as the authors described.
We have provided illustrative examples of negative and positive experimental results with the LFD in Figures 1C and 1D.
2. In figure 1B, the time on timer is not consistent with the illustration.
We have altered the times accordingly in Figure 1B, Steps 1, 2 and 3.
3. In figure 1D, what does the logo in the bottom left corner mean?
We have removed the BioRender logo.
4. Please supplement the results of repeated experiments in Table 1.
We are uncertain what is being requested here. As stated in the legend to Table 1, the data are the means of two replicate samples. We have adjusted the wording of the legend to make this clearer for the ELISA results.
5. Please improve the image quality of figure 2, and please supplement the results of repeated experiments and statistic results in figure 2.
Could the Reviewer please give further details of the improvements required as we believe this western blot image to be of good quality?
Furthermore, it is not possible to conduct statistical analysis on western blots.
6. Since the authors wanted to compare the LOD of ScedLFD test and ScedELISA, please unify the format of Figures 3 and 4.
We have now unified the format of Figure 4 to be consistent with Figure 3. Please see new Figure 4.
7. There are several critical papers related lateral flow assay should be discussed in the manuscript, such as Trends in Analytical Chemistry, 2024, 117641 and Biosensors 2023, 13 (3), 352.
We have now added the article by He et al. 2024 as reference 79. We do not believe the article by Liu et al. 2023 to be of relevance here as it describes a three-line LFIA.
Reviewer 2 Report
Comments and Suggestions for Authors
Dear authors,
Article is focused on a very important medical issue which is rapid diagnosis of potencially life-threatening infections caused by mold Scedosporium. There is no rapid antigen test currently available for the specific detection of Scedosporium species. Threfore the development of a lateral-flow device (ScedLFD) for the rapid (30-50 min) and sensitive (pmol/L) detection of a Scedosporium-specific biomarker is a very valuable outcome of your research.
Quality of Figure 2 could be improved much more. Can you replace it, please?
Please also provide a real-evidence of Sced-LFD test to confirm its excellence instead of Figure 1C and 1D, which are only illustrative.
Author Response
We thank Reviewer 2 for their constructive comments. We address their comments in bold below. We highlight relevant changes in yellow in the corrected manuscript attached.
1. Quality of Figure 2 could be improved much more. Can you replace it, please?
We are uncertain why the image is deemed to be of poor quality. Could the Reviewer please give further details of the improvements required?
Please also provide a real-evidence of Sced-LFD test to confirm its excellence instead of Figure 1C and 1D, which are only illustrative.
We have now provided real-evidence of negative and positive Sced-LFD test results in Figures 1C and 1D, respectively.
Round 2
Reviewer 1 Report
Comments and Suggestions for Authors
No comments any more.
Reviewer 2 Report
Comments and Suggestions for Authors
Dear authors,
thanks for improving quality of your paper, now I cen recommend it to be accepted.